# Elicitation of stakeholder viewpoints about medical cannabis research for pain management in critically ill ventilated patients: A Q-methodology study

Giulio DiDiodato[1,2]*, Samah Hassan[3], Kieran Cooley[3,4,5]

**1** Department of Health Research Methods, Evidence & Impact, McMaster University Medical Centre, Hamilton, Ontario, Canada, **2** Centre for Education & Research, Royal Victoria Regional Health Centre, Barrie, Ontario, Canada, **3** The Canadian College of Naturopathic Medicine, Toronto, Ontario, Canada, **4** Faculty of Health, Australian Research Center on Complementary and Integrative Medicine, University of Technology Sydney, Ultimo, Australia, **5** The Pacific College of Oriental Medicine, Chicago, Illinois, United States of America

\* didiodatog@rvh.on.ca

**Data Availability Statement:** All relevant data are within the manuscript and its Supporting Information files.

## Abstract

### Objectives

To determine acceptability of medical cannabis research in critically ill patients.

### Design

Q-methodology survey.

### Setting

Convenience sample of healthcare providers and the general public were recruited at an acute care community hospital in Ontario, Canada.

### Participants

In the first phase, 63 respondents provided 197 unique viewpoints in response to a topic statement about medical cannabis use in critically ill patients. Twenty-five viewpoints were selected for the q-sample. In the second phase, 99 respondents ranked these viewpoints according to an *a priori* quasi normal distribution ranging from +4 (most agree) to -4 (least agree). Factor analysis was combined with comments provided by survey respondents to label and describe the extracted factors.

### Results

The factor labels were *hoping and caring* (factor 1), *pragmatic progress* (factor 2), and *cautious/conservative and protectionist* (factor 3). Factor 1 describes a viewpoint of unequivocal support for medical cannabis research in this population with few caveats. Factor 2 describes a viewpoint of cautious support with a need to monitor for unintended adverse

**Funding:** This study was funded by Canopy Health Innovation in the form of an unrestricted educational grant awarded to GD, KC, and SH. In addition, the Royal Victoria Regional Health Centre Foundation provided funding to cover the publication costs associated with this study. The funders had no role in study design, data collection and analysis, decision to publish, or preparation of the manuscript.

**Competing interests:** The authors have read the journal's policy and have the following competing interests: Canopy Health Innovation provided funding for this study in the form of an unrestricted educational grant. This does not alter our adherence to PLOS ONE policies on sharing data and materials. There are no patents, products in development or marketed products associated with this research to declare.

effects. Factor 3 describes a viewpoint of ensuring that current analgosedation techniques are optimized before exposing patients to another potentially harmful drug.

## Conclusions

Using a q-methodology design, we were able to sample and describe the viewpoints that exist about medical cannabis research in critically ill patients. Three factors emerged that seemed to adequately describe the relative ranking of q-statements by the majority of respondents. Combining the distinguishing statements along with respondent comments allowed us to determine that the majority support medical cannabis research in critically ill patients.

## Introduction

Life-sustaining therapies, such as invasive mechanical ventilation, are commonly employed in patients admitted to an intensive care unit (ICU) with life-threatening illnesses. The majority of these patients require pain control and sedation to facilitate their care but also to alleviate suffering directly related to their illness [1]. Opioids represent the primary pharmacologic agents used to manage pain and sedation in the ICU. This is referred to as analgosedation, analgesia-mediated sedation. Ideally, critically ill patients should be exposed to the minimum doses of opioids that relieves their suffering, facilitates their care, and exposes them to the minimum number of both short- and long-term consequences and complications. Unfortunately, there is no *a priori* established dose range of opioids that fulfils these requirements for every patient given the subjective nature of pain and suffering [2]. The current strategies to manage pain in critically ill patients depend on using validated pain scoring systems and frequent clinical assessment by ICU nurses to guide the titration of opioids and other pharmacologic therapies used for analgosedation [3]. Despite these considerable efforts, many patients still receive either too little or too much opioids during their ICU admission, either of which may have many negative consequences for patients. Too little opioids expose patients to unrelieved pain that may adversely affect their psychological well-being, cause them to resist care provided by health care providers, put them at risk of self-harm from events such as self-extubation, and may activate pathologic stress responses resulting in events such as gastrointestinal bleeding or hypertension. Too much opioids exposes patients to excess days on a ventilator due to respiratory depression, altered levels of alertness and severe delirium, and the development of tolerance to opioids requiring ever-increasing amounts to manage pain that may lead to the risk of a withdrawal syndrome and chronic opioid tolerance.

Medical cannabis has been shown to be an effective therapy in treating chronic pain [4]. The evidence for efficacy of medical cannabis use to alleviate other pain syndromes is currently lacking [5]. There are no clinical trials demonstrating the efficacy of using medical cannabis as an adjunct to analgosedation as either an opioid-sparing agent or to alleviate the psychological complications associated with critical illness such as delirium or post-traumatic stress disorder (PTSD). A recent search of *ClinicalTrials.com* using the search terms "Medical cannabis", "Medical marijuana", "Medicinal cannabis", "Marijuana treatment", and "Medicinal marijuana" yielded 167 studies, none of which involved critically ill patients (https://clinicaltrials. gov/ct2/results/details?term=Medical+cannabis). Despite this lack of evidence, some pre-clinical studies suggest that medical cannabis may have an important role in modulating the inflammatory response, an important potentiator of both pain and illness severity in critically

ill patients [6, 7]. In addition, preliminary studies suggest a positive effect of medical cannabis on PTSD-related symptoms such as nightmares [8]. PTSD is a common long-term consequence of critical illness as part of a syndrome referred to as post-intensive care syndrome (PICS) that can affect up to 30% to 50% of ICU survivors [9]. These potential benefits of medical cannabis, along with its recent legalization and *lower* risk profile compared to opioids [10], make it a candidate therapy for investigation in critically ill patients at high risk of opioid tolerance and PICS.

Apart from the clinical rationale for medical cannabis use in critically ill patients, there have been no studies exploring the viewpoints of both the public and health care providers on the potential role of medical cannabis in these patients. To date, almost all cannabis products approved for medical use demonstrate efficacy in patients with terminal or refractory health conditions such as chronic cancer pain and epilepsy [4]. Given that there are alternative strategies to analgosedation and the risks from medical cannabis use may not be as benign as previously believed [11–13], it remains uncertain whether there would be both public and health care provider support for medical cannabis research in this complex patient population.

Q-methodology is a mixed method study design used to describe different viewpoints that may exist about any topic [14]. Participants are provided with a topic statement to review. This statement is rarely neutral as it is intended to elicit divergent and strong opinions. Participants are then asked to provide their viewpoints in response to this topic statement in 2 different phases. The first phase involves generating a q-set, a theoretic population of all viewpoints that may exist. These viewpoints may emerge from many different sources such as journal articles, expert opinion, popular media or stakeholders. Once collected, a strategic sampling of the q-set is completed to ensure that all unique and relevant viewpoints are represented in a q-sample. In the second phase, stakeholders are asked to rank the q-sample statements according to their agreement using an *a priori* quasi normal distribution, a q-sort. This relative ranking creates a unique q-sort pattern for each respondent. In addition to the q-sort, stakeholders are asked to provide qualitative comments about why they ranked statements in the most agree, least agree and neutral categories. The q-sorts are then analysed using factor analyses techniques, and only those factors with significant loadings on the q-sorts are retained. These factors are labelled and described using both distinguishing q-sample statements and the comments provided by stakeholders. These factors represent latent viewpoints about the topic of interest.

We designed a q-methodology study to determine if health care providers and the public would support medical cannabis research in critically ill ventilated patients, and demonstrated a favourable response from both groups in their support of this research activity.

## Materials and methods

The study was approved by the Royal Victoria Regional Health Centre Research Ethics Board (REB#R18-010) on September 26, 2018. Consent was obtained from each study participant before entering the survey.

### Q-methodology design

We used a q-methodology study design to systematically explore and describe the range of viewpoints about medical cannabis research in critically ill ventilated patients among relevant stakeholders [15]. Our study was limited to stakeholders who might be directly affected by the use of medical cannabis in critically ill patients. The investigators created a topic statement about the use of medical cannabis in critically ill patients using their content expertise in pain medicine (SH), naturopathic medicine (KC) and critical care (GD) (S1 Appendix, Topic

Statement). This topic statement was used in both phases of the q-methodology study. The survey participants for both phases were sampled from both the lay public and health care providers located at the Royal Victoria Regional Health Centre in Barrie, Ontario by using a non-probability strategy. The survey participants were strategically sampled to ensure high potential for broad, discrepant and extensive sets of viewpoints were represented in the q-set and q-sample phases. Survey respondents who participated in phase 1 were asked not to participate in phase 2. Survey participants that had critical care experience and that were critical care-naïve were sampled from both the lay public and health care provider groups. The usual total sample size target for q-set development is between 40 to 60 survey participants [15]. All the statements from phase 1 were reviewed by all three investigators using an *informal* approach to finalize the statements that were included in the q-set. This informal approach involved each of the 3 investigators independently reviewing the entire set of q-set statements. The investigators independently grouped similar statements into themes, and created q-set statements that were representative of these groupings. Once this was completed, the investigators independently ranked the q-set statements in order from most relevant/preferred for inclusion to least relevant/preferred for inclusion. All the investigators attempted to rank and include the q-set statements in such a way as to ensure their final q-sort statements represented the most unique and diverse viewpoints. Once the investigators each completed their independent review of the q-set statements, they convened to review each other's q-set statements' rankings. Through an iterative process, the final q-sort statements were chosen if there was unanimous agreement among the 3 investigators for inclusion. The investigators also used their own clinical expertise to supplement the final q-sort statements to ensure a sufficiently broad and comprehensive set of viewpoints were included in the final q-sort. From this process, a final list of 25 statements was chosen by the investigators to create the q-sample (S1 Appendix in Table 1). In phase 2, participants ranked these q-sample statements according to their relative agreement or disagreement using a forced distribution (Fig 1), with each survey respondent creating their own unique q-statement ranking pattern, q-sort. The usual total sample size target for q-sort is also between 40 to 60 survey participants.

Further qualitative information was requested after the q-sort by asking survey participants to explain why they had ranked statements at either extreme of the quasi normal distribution, why they had ranked statements as neutral and their overall viewpoints about the topic statement. Completed q-sorts were analyzed using principal factor analysis, a quantitative data reduction technique, to identify survey respondents with highly correlated q-sort patterns [16]. These intercorrelated survey participants were considered to share similar viewpoints about the topic statement, and these viewpoints are termed *factors*. Factors represent the observable viewpoints of latent perspectives. The output from this analysis are 'idealized' q-sorts with characteristic patterns for each factor. Survey participants' q-sorts were variably explained by one or a combination of these factors. The extent that each participant's q-sort is explained by these factors is termed a factor load. The factors that explain most of the variation across $\geq 2$ q-sorts were retained. The final output of the analysis was a *descriptive labelling* of each factor that emerged by combining the q-sorts, along with those statements that distinguish one factor from another and was further supported by the qualitative statements provided by participants at the end of the q-sort.

## Data collection and analysis

Both phases of the q-methodology study were programmed in REDCap© (https://www.project-redcap.org), a Personal Health Information Protection Act-compliant database stored at the Royal Victoria Regional Health Centre (Access to the q survey phase 1 and phase 2

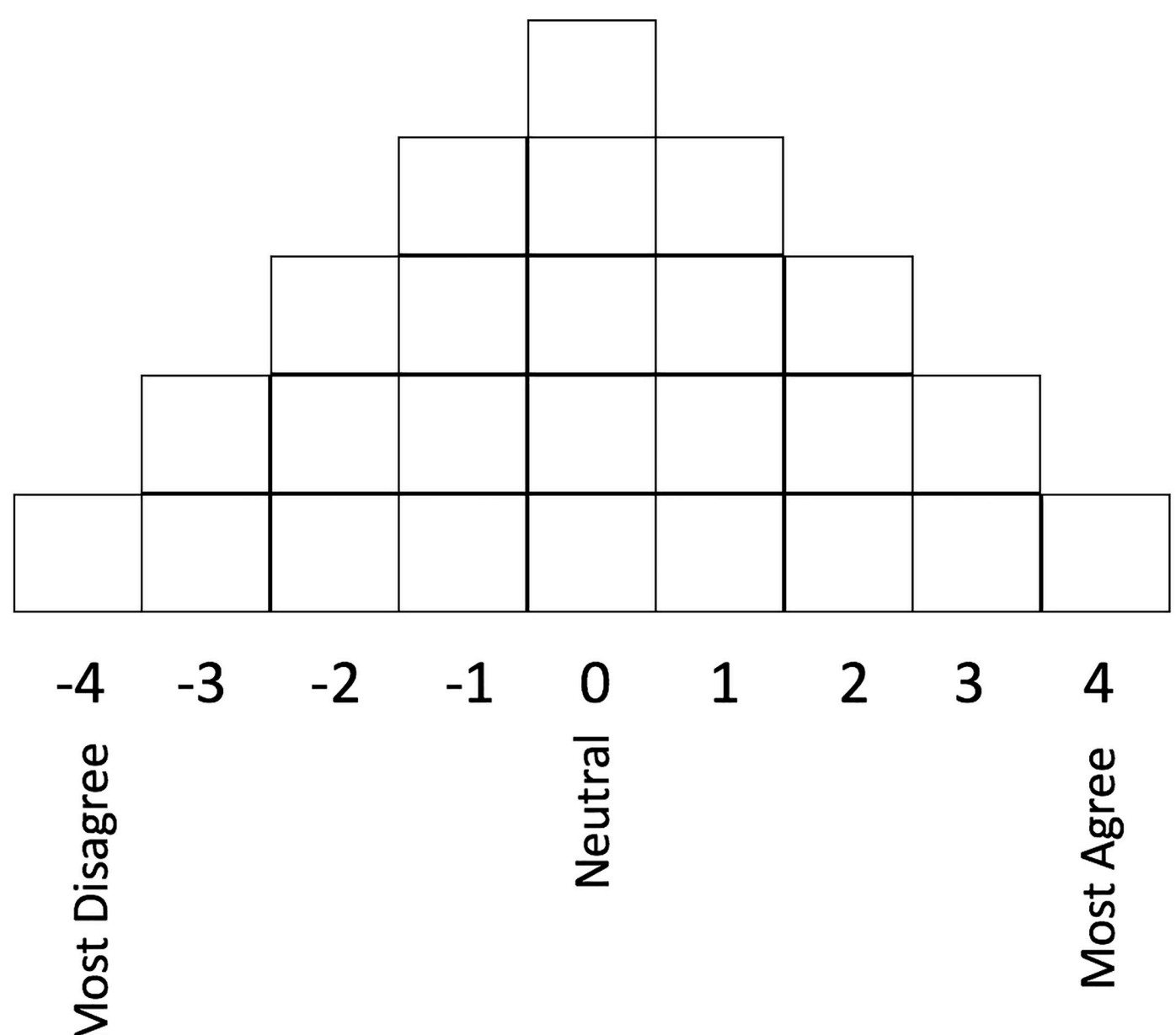

**Fig 1. Quasi normal distribution used to force ranking of q-statements by participants.**

templates for review is available upon request). After obtaining Research Ethics Board approval on September 26, 2018 (REB #R18-010), a link for the phase 1 q-set survey was distributed to all intensive care unit (ICU) staff email addresses along with a request for participation. Snowball sampling of other hospital staff was done by asking ICU staff to forward the request for participation emails along with the survey link to other hospital staff they felt would be interested in learning more about the study and potentially participating. In addition, a study brochure explaining the rationale, the research question, the requirements of participation and the study survey link was made available to families and patients in the ICU by the ICU charge nurse during daily inter-disciplinary rounds. In addition, volunteer high school students approached potential participants in the hospital's common areas, such as the

cafeteria, to explain the rationale for the study and provide interested participants with a study brochure to support lay public recruitment. The study brochures were also advertised at all hospital entrances to support lay public recruitment. Participants were asked to provide informed consent online prior to participation. The phase 1 q-set survey was open from October 1, 2018 to November 30, 2018 which was defined *a priori*. Once completed, the investigators reviewed all the participants' statements and created a final q-set from which the q-sample statements were selected. The phase 2 q-sample survey was similarly distributed. The q-sample survey was open from December 10, 2018 to February 28, 2019. All quantitative analyses were conducted using the *qfactor* command in STATA 15/MP [17]. The iterated principal factor method was used to extract from 2 to 5 factors. Promax oblique rotation was used after factor extraction for further data reduction and refinement of factor groupings. The regression method was used to estimate factor z-scores. Distinguishing statements between factors were identified using a Cohen's d-value $\geq 0.8$. All participants' comments regarding their rationale for ranking were reviewed by all three investigators. Using these free-hand comments to supplement the distinguishing factor statements, a final label and description for each of the retained factors was developed. To label the factors, the investigators identified strongly differentiating statements using the following criteria: those with a $\geq |2|$ difference in ranking score from other factors along with having an extreme ranking score value (either +4 or -4 or zero for neutral). The investigators supplemented these with the supportive qualitative comments to assist with final factor labelling and description.

## Results

In phase 1, there were 105 participants who consented to the survey, with 63 respondents submitting their viewpoints. The distribution of lay public and health care provider participants with and without critical care experience was unbalanced (Table 1).

From these 197 unique viewpoints, the investigators created a q-set consisting of 70 statements. From these 70 statements, 25 final viewpoints were sampled to create the q-sample (S1 Appendix in Table 1).

In phase 2, there were 253 participants who consented, with 118 respondents submitting their q-sorts. After cleaning the database, 99 q-sorts were included in the final analysis as 19 surveys had missing data (Table 2). There was a much more balanced representation across public and health care provider groups.

For this study, we ultimately extracted 3 factors using the rationale that there were likely to be 2 groups defined by opposing viewpoints on critical issues, and a third group that was defined as being either supportive or against depending on the circumstances. The factor

**Table 1. Number and distribution of statements among health care providers and the public from phase 1.**

| Respondent Type[1] | Respondents (Total number) | Age group (Total number) | | | Statements (Total number) |
|---|---|---|---|---|---|
| | | Under 30 | 30–50 | Over 50 | |
| HCP ICU+ | 47 | 8 | 28 | 11 | 147 |
| HCP ICU- | 11 | 3 | 7 | 1 | 33 |
| Public ICU+ | 2 | 0 | 1 | 1 | 6 |
| Public ICU- | 3 | 1 | 0 | 2 | 11 |
| Total | 63 | 12 | 36 | 15 | 197 |

[1] HCP ICU+ = any health care provider whose primary clinical care area is the ICU (this could include nurses, physicians, dieticians, physiotherapists, pharmacists, medical residents, etc.); HCP ICU- = any health care provider whose primary clinical care area is outside the ICU; Public ICU+ = lay public who have family or friends admitted to the ICU; Public ICU- = lay public who have never had family or friends admitted to the ICU.

**Table 2. Number and distribution of completed q-sorts among health care providers and the public from phase 2.**

| Respondent Type | Respondents (Total number) | Age group (Total number) | | |
|---|---|---|---|---|
| | | Under 30 | 30–50 | Over 50 |
| HCP ICU[+] | 31 | 7 | 20 | 4 |
| HCP ICU[-] | 29 | 5 | 20 | 4 |
| Public ICU[+] | 9 | 5 | 4 | 0 |
| Public ICU[-] | 20 | 15 | 4 | 1 |
| Total | 99 | 32 | 48 | 9 |

extraction resulted in 3 factors with their own unique q-sort pattern that serves as the "best-estimate" or "average" of the statement ranking configuration for that factor (Fig 2).

As seen in Fig 2, the 3 factors have considerable overlapping patterns, differing mostly on degree of ranking as opposed to having opposing rankings. When one looks at how much of the variance in q-sort patterns was explained by each factor, Factor 1 accounts for the majority of the variance (Table 3), with 48 participants' q-sorts being best described by Factor 1.

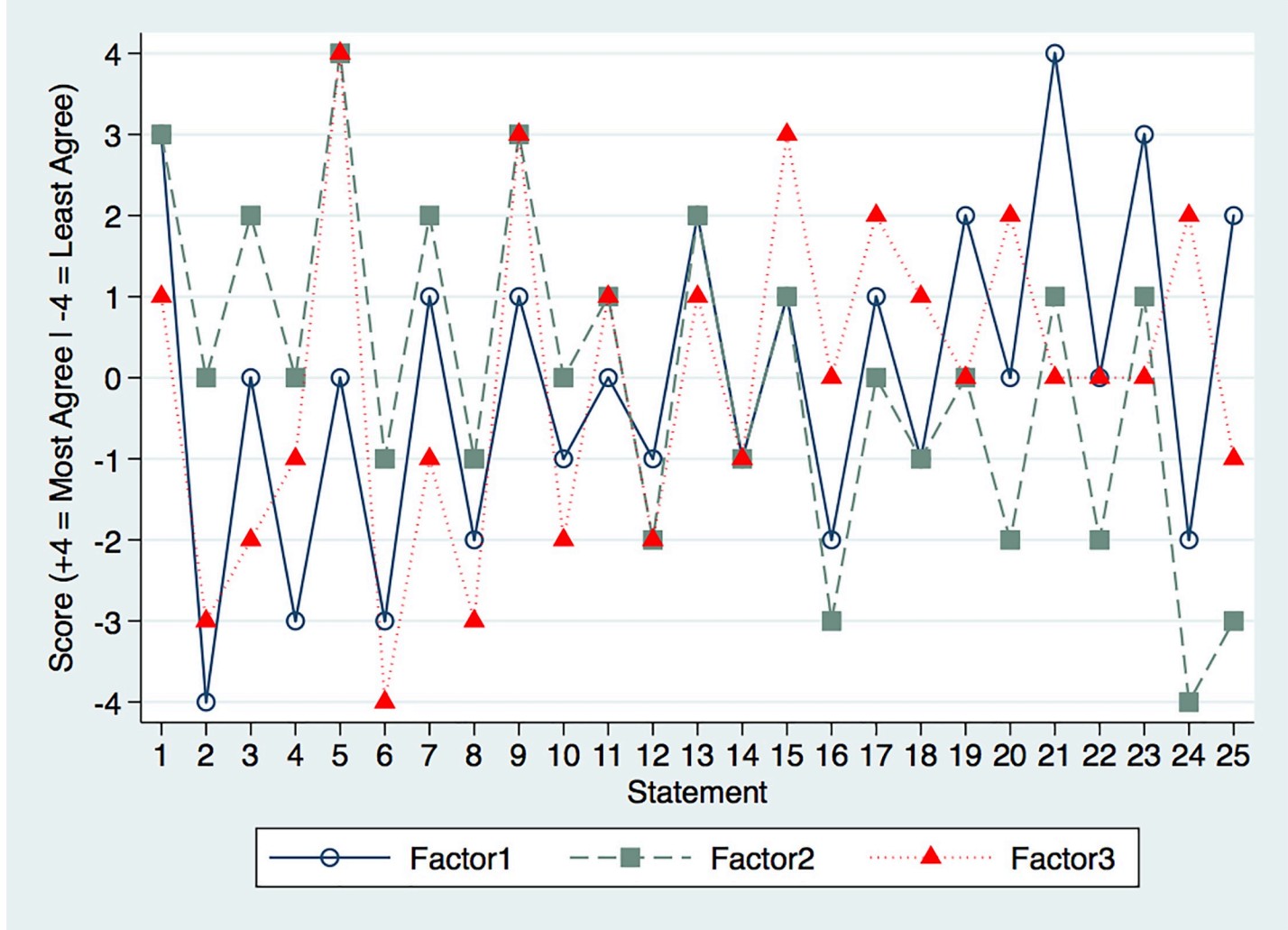

**Fig 2. Q-sort patterns of 3 uniquely identified factors.**

**Table 3. Variance in q-sort patterns due to Factors 1, 2 and 3.**

| Factor | Eigenvalue (= Variance) | Proportion (of Total Variance) | Q-sorts (Total number) |
|--------|-------------------------|--------------------------------|------------------------|
| 1 | 47.56 | 0.81 | 48 |
| 2 | 6.39 | 0.11 | 12 |
| 3 | 4.76 | 0.08 | 7 |

The correlation between factor patterns was low, with all correlation coefficients ranging from 0.3 to 0.49. Factor loadings reflect correlations between the observed q-sort patterns and their respective factor patterns, where the (factor loading)$^2$ = variance of the q-sort pattern attributable to that factor (S1 Appendix in Table 2).

The distribution of factor loadings between the public and health care providers did demonstrate some variability. There were no differences between the distribution of either factor 1 ($\chi^2$ (3) = 4.97, p = 0.174) or factor 3 ($\chi^2$ (3) = 6.01, p = 0.111), but there was a difference between these groups for factor 2 ($\chi^2$ (3) = 16.89, p = 0.001) with the majority of the public (both with and without ICU experience) (10/20 participants) being best described by this factor. There were significant differences in the distribution of factors between different age groups: factor 1 ($\chi^2$ (3) = 23.01, p<0.001), factor 2 ($\chi^2$ (3) = 9.32, p = 0.025), and factor 3 ($\chi^2$ (3) = 11.29, p = 0.010). The majority of under 30 year olds were best described by factors 2 and 3 (16/32), while factor 1 best described those between 30 and 50 years old (35/48), and those over 50 years old were best described by both factors 1 and 2.

Distinguishing q-sample viewpoints are those statements that are ranked uniquely for one factor compared to the others that serve to help label the factor (S1 Appendix in Table 3). In this study, distinguishing statements were defined as demonstrating differences ≥ 2 between average factor scores. By this criteria, factor 1 had 6, factor 2 had 3 and factor 3 had 4 distinguishing statements.

As opposed to distinguishing statements, there were also consensus q-sample viewpoints. These elicited similar rankings across all factors (S1 Appendix in Table 4). There were 7 consensus statements.

The final factor labelling *was Hoping and Caring (Factor 1)*, *Pragmatic Progress* (Factor 2) *and Cautious/Conservative and Protectionist (Factor 3)*. A description of these factors and an example of their supporting statements are provided in the supplement (S1 Appendix in Table 5).

## Discussion

The perception of medical cannabis and its use as a medicinal therapy is changing globally as more countries have legalized recreational marijuana use and liberated prescribing restrictions for physicians [18, 19]. Even the World Health Organization is reconsidering its policies around medical cannabis, and is set to vote on rescheduling cannabis from its current Schedule IV classification that restricts its use as a legitimate medical therapeutic [20]. These changing policies and attitudes have led to an upsurge in cannabis research [21]. It is in this emerging climate of increasing public and provider acceptance of both recreational and medical cannabis that we undertook this study to examine the feasibility of studying medical cannabis as an adjunct to analgosedation in critically ill patients.

In this study examining viewpoints about cannabis research in critically ill ventilated patients, we demonstrated that general consensus among sampled health care providers and the public exists supporting this research activity. By using a q-methodology study design, 3 unique factors were identified that were used to adequately describe the q-sort patterns of

more than 67% of the survey participants. The factors described viewpoints that ranged from being fully supportive of cannabis research without caveats to supporting a cautious research approach that was limited in scope and patient eligibility. This study is the first to use a q-methodology approach to elicit viewpoints about cannabis research among stakeholders. Previous studies have demonstrated varying levels of public and health care provider support for medical cannabis use that is dependent on many contextual factors that include clinical indication and knowledge and beliefs about the benefits and harms associated with medical cannabis use [22–24]. By using this approach, we were able to reduce the complexity of this topic into a practicable number of shared viewpoints. In addition, the results of the study also provide guidance on whom to include and exclude in medical cannabis research, what outcomes should be targeted for impact and who should be involved in the design, financing, conduct and analysis of the research thus providing a novel technique to include stakeholders in planning future research studies. For example, at risk populations such as pregnant women, those with substance abuse disorders or mental health patients were identified as being groups that should be excluded from inclusion in any subsequent studies due to potential risks associated with medical cannabis use. Many stakeholders identified that long-term outcomes such as medical cannabis abuse disorder as a negative consequence of enrolment should also be monitored as a safety signal. In addition, the majority of stakeholders agreed that medical cannabis producers and companies who are funders of any subsequent studies should be completely removed from the design, conduct, analyses and reporting of the study results.

The major limitation of the study is the external validity of the results. While we attempted to create a comprehensive set of viewpoints to be included in the q-sort, there was a statement imbalance in creating the concourse between the lay public and health care provider groups. In addition, the method of choosing the final q-statements from the q-set generated in phase 41 of the study was left up to the 3 investigators. While this selection process required agreement between all 3 investigators for final inclusion and all 3 investigators had very different backgrounds, there may have been some selection bias in choosing the final q-sort statements that could have affected the final factors that emerged from the study, Having said this, every q-study is intrinsically limited by the process that must be implemented to choose the final q-sort statements and our study was consistent in its approach to that used in other q-studies. However, q methodology is robust to small sample sizes as it doesn't seek to determine the distribution of these viewpoints in the population, only the number of different viewpoints that might exist. Analysis of the distribution of these groups to different factors confirmed that while there were important differences, these differences were mostly in degree of support for medical cannabis research as opposed to contrasting opinions (See Description in S1 Appendix in Table 5). In addition, approximately 32% of survey participant's q-sort patterns were not described by the 3 factors described in this study, suggesting there may exist significant heterogeneity of viewpoints beyond those we described. A review of their comments, however, did not reveal strong viewpoints against supporting cannabis research but rather neutrality about the subject.

## Conclusions

*Hoping and caring, pragmatic progress* and *cautious and protectionist* viewpoints emerged as the three dominant themes among stakeholders. There was significant overlap in these viewpoints for their support of research involving the use of medical cannabis as an analgosedative agent in critically ill ventilated patients. The difference between these viewpoints was in the degree of support with the *hoping and caring* unequivocal in their support while the *cautious and protectionist* imploring optimization of current therapies before the introduction of

another potentially harmful drug. Given this stakeholder input, we belive there is sufficient support for proceeding with medical cannabis research as an adjunct to analgosedation in critically ill ventilated patients. The next steps involve planning for a pilot study to investigate the doses and formulations of enteral medical cannabis that could be safely administered to critically ill patients while measuring outcomes related to opioid-sparing effects and ameliorating the incidence and severity of other ICU-related complications such as acute delirium and PICS.

## Supporting information

**S1 Appendix. Topic statement.**
(DOCX)

**S1 Data.**
(XLSX)

## Acknowledgments

We would like to thank the Royal Victoria Regional Health Centre Foundation for their support of the publication costs.

## Author Contributions

**Conceptualization:** Giulio DiDiodato, Samah Hassan.

**Data curation:** Giulio DiDiodato, Samah Hassan, Kieran Cooley.

**Formal analysis:** Giulio DiDiodato, Samah Hassan, Kieran Cooley.

**Funding acquisition:** Giulio DiDiodato, Kieran Cooley.

**Investigation:** Giulio DiDiodato, Samah Hassan, Kieran Cooley.

**Methodology:** Giulio DiDiodato, Samah Hassan, Kieran Cooley.

**Project administration:** Giulio DiDiodato, Samah Hassan.

**Resources:** Giulio DiDiodato, Kieran Cooley.

**Software:** Giulio DiDiodato.

**Supervision:** Giulio DiDiodato.

**Validation:** Giulio DiDiodato, Samah Hassan, Kieran Cooley.

**Writing – original draft:** Giulio DiDiodato, Samah Hassan, Kieran Cooley.

**Writing – review & editing:** Giulio DiDiodato, Samah Hassan, Kieran Cooley.

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
