## [Decision Letter · Decision Letter 0]

12 Nov 2020

PONE-D-20-16468

Elicitation of stakeholder viewpoints about medical cannabis research for pain management in critically ill ventilated patients: A Q-methodology study

PLOS ONE

Dear Dr. DiDiodato,

Thank you for submitting your manuscript to PLOS ONE. After careful consideration, we feel that it has merit but does not fully meet PLOS ONE’s publication criteria as it currently stands. Therefore, we invite you to submit a revised version of the manuscript that addresses the points raised during the review process.

The manuscript has been evaluated by two reviewers, and their comments are available below. You will see the reviewers have commented on the importance of your work. However, they have also raised a number of concerns that should be addressed before the manuscript can be further considered for publication.

The key concerns noted by the reviewers relate to the methods and results. Specifically, the reviewers requested further background and justification for the q-methodology. Additionally, the reviewers noted that the presentation of the results in the main manuscript could be elaborated. These issues have limitations for the interpretation of the results and should be explored.

We look forward to receiving your revised manuscript.

Kind regards,

Danielle Poole

Staff Editor

PLOS ONE

Journal Requirements:

2. Please do not include funding sources in the Acknowledgments or anywhere else in the manuscript file. Funding information should only be entered in the financial disclosure section of the submission system. https://journals.plos.org/plosone/s/submission-guidelines#loc-acknowledgments

3.Thank you for stating the following in the Financial Disclosure section:

[The authors received an unrestricted educational grant from Canopy Health Innovation.  Canopy Health Innovation had no role in the study design, data collection and analysis, decision to publish, or preparation of the manuscript. ]. 

We note that you received funding from a commercial source: [Canopy Health Innovation]

4.We note that you have indicated that data from this study are available upon request. PLOS only allows data to be available upon request if there are legal or ethical restrictions on sharing data publicly. For information on unacceptable data access restrictions, please see http://journals.plos.org/plosone/s/data-availability#loc-unacceptable-data-access-restrictions.

5. Please ensure that you refer to Figure 1 in your text as, if accepted, production will need this reference to link the reader to the figure.

Reviewers' comments:

Reviewer's Responses to Questions

**Comments to the Author**

1. Is the manuscript technically sound, and do the data support the conclusions?

Reviewer #1: No

Reviewer #2: Yes

2. Has the statistical analysis been performed appropriately and rigorously? 

Reviewer #1: No

Reviewer #2: Yes

3. Have the authors made all data underlying the findings in their manuscript fully available?

Reviewer #1: No

Reviewer #2: Yes

4. Is the manuscript presented in an intelligible fashion and written in standard English?

Reviewer #1: Yes

Reviewer #2: Yes

5. Review Comments to the Author

Reviewer #1: The topic of MC research in critically ill patients is important and novel. The present study uses an interesting and rich methodology to sample and describe the viewpoints that exist about medical cannabis research in critically ill patients. I think some work is needed in order to better describe the results and implications of this study.

The methodology used here may be new for many readers. While it is described well it would be beneficial to add some background as a justification for the choice of this method and explain why this method was used instead of others, e.g. traditional qualitative methods and Delphi methods.

It is stated that Completed q-sorts were analyzed using principal factor analysis, a quantitative data reduction technique, to identify survey respondents with highly correlated q-sort patterns. I am no expert in factor analyses so I may be wrong. However, I thought that factor analyses examines items that are part of the same constructs. It does not help distinguish different types of survey respondents.

In order to compare factors across the 2 groups the authors should first run measurement invariance testing to assure that the assumption of measurement invariance was met.

The final factor labelling, and thus their actual meaning is described only at the very end of the result section. And also very briefly. I would suggest describing the meaning of the factors in more detail and also lifting this further up in order to make the rest of the analyses more meaningful to the reader. So when the author speaks of group difference in certain factors, it would be more meaningful if the reader already knew what the meaning of the factor in question is. To create more comprehension it would also be beneficial to use the label of each factor when describing the results as opposed to just the abstract “factor 1” label.

The limitation of the method is very brief. It is noted that this study shows there is support for research on MC in critically ill therefore next step is to do research. I wonder if there may be other elements of concern (over and above the opinion of a convenience sample) that needs to be included before research can proceed. A more critical assessment of how the results of this study can be used in practice should be developed. The results are based on a convenience sample and the authors note this clearly. However, this fact may have some bearing on the significance of the results and this should be considered more carefully.

Reviewer #2: 1. The introduction is good, but long. Much of it can be moved to discussion.

2. Most of the actionable result is included in the appendix. I would encourage the authors to move that into the main manuscript.

3. Encourage to not include inferential statements in the results. Move these to discussion. Example- lines 238- 242 and 245.

4. Under discussion- in lines 297- 301 authors mention that the study provides guidance on various aspects related to cannabis research, however, it is not clear what exactly is that guidance. This needs to be further elaborated and discussed as it is the most important part of the results.

5. The authors make strong conclusion statement (which one long sentence and should be broken down for easy readability). However, i did not see this conclusion evolve in the discussion.

6. PLOS authors have the option to publish the peer review history of their article (what does this mean?). If published, this will include your full peer review and any attached files.

Reviewer #1: No

Reviewer #2: **Yes: **Vinita Singh

---

## [Author Response · Author response to Decision Letter 0]

23 Dec 2020

PONE-D-20-16468

Elicitation of stakeholder viewpoints about medical cannabis research for pain management in critically ill ventilated patients: A Q-methodology study

PLOS ONE

Reviewer#1

Thank you for taking the time to review our manuscript.

1) Why did we choose Q-methodology over other qualitative techniques?

One of our group (GD) has a quantitative statistical background, and Q-methodology has a strong quantitative influence unlike the other qualitative methods. We reviewed several papers that used Q-methodology and we could not find a single example where the authors tried to justify their use of this technique versus using pure qualitative approaches as you’ve mentioned. We believe the reason is that Q-methodology is both a qualitative and quantitative approach to viewpoint solicitation and so is different enough from purely qualitative techniques that it would not be necessarily appropriate to discuss it only in terms of a qualitative approach.

2) Factor analysis question.

Q-methodology uses factor analysis to group overall patterns of statement rankings about a topic. It is the common viewpoints and opinions that emerge that are uncovered by the factor analysis, whereas traditional factor analysis looks for correlations between characteristics such as how responses to different items in a questionnaire can be grouped together.

3) Measurement invariance

We are not comparing factors across any groups; we are looking for patterns in the overall ranking of statements. Traditional factor analysis is not used in Q-methodology, so measurement invariance is not part of the quantitative analysis. We did provide a reference (15) explaining the quantitative approach to Q-methodology that we feel help clarify the approach for those who are less familiar with this method.

4) Final factor labelling

In Q-methodology, the factor labelling is the final step and the product of the technique. We cannot move it forward as it emerges after the quantitative analysis and a review of all the qualitative statements supporting the ranking patterns that emerged as a result of the quantitative analysis.

5) Method limitation

We did provide additional limitations beyond that described by the reviewer that included the imbalance in the first phase between the lay public and healthcare providers, and also that the final 3 viewpoints that emerged did not include the patterns elicited in 32% of the remaining participants. While we have used the term ‘convenience’ sample, it is not to be interpreted in the context of survey sampling as Q-methodology is not a survey technique and is not limited by sample representativeness. Instead, Q-methodology seeks to ensure that sufficient stakeholders are included whose viewpoints should span the imaginable viewpoints that might exist on any topic of interest. The ‘convenience’ part of our sample was that we solicited these viewpoints from stakeholders all located from the same hospital. This would not be considered to be a limitation in Q-methodology.

Reviewer#2

Thank you for taking the time to review our manuscript

1) Introduction too long

We wanted to ensure we provided sufficient background for the Q-methodology approach as we recognize this method is unfamiliar to many readers. We do feel that it is comprehensive in both providing the background on analgosedation in critically ill patients, cannabis use and Q-methodology in a way that provides the reader with enough information to understand the design and results.

2) Appendix materials

The output of q-methodology is quite extensive and sometimes difficult to understand for unfamiliar readers, so we decided to be as comprehensive as possible and include almost all of it both the results and the appendix. We felt we had to use an extensive appendix to support the qualitative portions of the method.

3) Inferential statements in results lines 238, 242, 245

These were removed and replaced in discussion

4) Guidance on important aspects related to cannabis researchers

We have updated that section of the discussion to highlight some important inputs into design, conduct, analysis and reporting commented on by the stakeholders in their subjective statements.

5) Conclusion

We have modified the conclusion to better reflect the reviewer’s comments

---

## [Decision Letter · Decision Letter 1]

25 Jan 2021

PONE-D-20-16468R1

Elicitation of stakeholder viewpoints about medical cannabis research for pain management in critically ill ventilated patients: A Q-methodology study

PLOS ONE

Dear Dr. Giulio DiDiodato

Thank you for submitting your manuscript to PLOS ONE. After careful consideration, we feel that it has merit but does not fully meet PLOS ONE’s publication criteria as it currently stands. Therefore, we invite you to submit a revised version of the manuscript that addresses the points raised during the review process.

We look forward to receiving your revised manuscript.

Kind regards,

Saeed Ahmed, MD

Academic Editor

PLOS ONE

Additional Editor Comments (if provided):

This study is aimed at describing the viewpoints of stakeholders towards conducting research on the use of Cannabis for Sedation and Pain management in Critically ill patients. The study uses Q- Methodology, a mixed-methodology design, to identify the major viewpoints that emerge from survey responses to statements around the objectives, design, risk and benefits, and, funding related to such research activities.

The study is well designed and the authors have implemented appropriate methods and statistical analyses.

The results are presented and discussed with clarity.

1) Given that all the viewpoints were obtained from a sample population from one hospital - How generalizable are their conclusions? Can these conclusion be generalized to other geographical locations which may have differing attitudes towards the use of Medical Cannabis in Critically ill patients. The authors' comment addressing this will be valuable.

Reviewers' comments:

Reviewer's Responses to Questions

**Comments to the Author**

1. If the authors have adequately addressed your comments raised in a previous round of review and you feel that this manuscript is now acceptable for publication, you may indicate that here to bypass the “Comments to the Author” section, enter your conflict of interest statement in the “Confidential to Editor” section, and submit your "Accept" recommendation.

Reviewer #3: (No Response)

Reviewer #4: All comments have been addressed

2. Is the manuscript technically sound, and do the data support the conclusions?

Reviewer #3: No

Reviewer #4: Yes

3. Has the statistical analysis been performed appropriately and rigorously? 

Reviewer #3: Yes

Reviewer #4: Yes

4. Have the authors made all data underlying the findings in their manuscript fully available?

Reviewer #3: No

Reviewer #4: Yes

5. Is the manuscript presented in an intelligible fashion and written in standard English?

Reviewer #3: Yes

Reviewer #4: Yes

6. Review Comments to the Author

Reviewer #3: Thanks for the opprtunity to review the manusript. The study method utilized seems somewhat novel and concentrates on an interesting area that needs exploration.

However, the study is challenged by significant methodological issues.

Utilizing a strategic sampling method and using an “informal approach” to finalize the “statements” included in the q- test indicates that the statement's selection was subjective to bias. The authors should mention this limitation in the discussion part and also they should comment on what basis the authors strategically choose the survey participants?

The authors should also describe the statements that were chosen for the survey. The authors should mention how they excluded the possibility of the same person being examined in both phases of the survey. The possibility of double-counting increases as they have utilized a snowballing technique to find the study participants.

There is a certain lack of clarity about the abbreviations and designations used by the authors. For example, at first, the authors have mentioned that phase 1 q set was distributed to all “intensive care unit staff” with snowball sampling of other hospital staff. They also mentioned that the face 2 q simple survey was “similarly distributed”. However, later on in the results section and on the tables, the authors have changed the terminology of "intensive care unit staff" with "healthcare providers". We need to keep in mind that staff members can be used to define various designations in a healthcare system, whereas providers are used only for doctors, nurse practitioners, and other very limited sets of professionals. The perspectives of healthcare providers can be very different from that of a layperson. Clubbing the healthcare providers and laypersons in the same group and asking for their opinion on a scientific research topic (whether medical cannabis research should be carried out or not and concluding on whom should be included in those studies) seems to be flawed. Healthcare providers and laypersons have wide differences in their knowledge and expertise in research and knowledge about cannabis.

A better study design could have been a comparison between healthcare providers viewpoint with that of laypersons (the authors may not agree, but this is the reviewer’s viewpoint). Similarly, it is hard to find if the authors have mentioned the full form of the abbreviation used in the tablets such as HCP ICU+, HCP ICU-, or Public ICU+ and Public ICU -.

The authors also should elaborate on factor 1, factor 2 factor 3 other than simply saying “hoping and caring”, “pragmatic progress” and “cautious/conservative and protectionist”.

The authors should also define what they mean by the “stakeholders” of medical cannabis research. This is particularly important because, the stakeholders of cannabis research does not only include the intensive care providers and family members but also includes, a wide range of other individuals such as government agencies, pain medicine specialists, substance users, psychiatrist and behavioral specialist that the authors have not included in this survey. The authors seem to have used the term stakeholders of medical cannabis research in a very simplistic way.

The authors also mentioned that the results of the study also provided guidance on "whom to include and exclude in medical cannabis research" and "what outcome should be targeted for impact" and "who should be involved in the design, financing, conduct an analysis of the research". Unfortunately, this study does not seem to really conclude the above based on the study method hat the authors have used. Can the author comment on how they concluded the above crucial issues based on a survey involving the “viewpoint (which may or may not have any scientific basis)” of laypersons and ICU providers? The authors should also comment on how did this survey help them to come to a conclusion that "at-risk populations such as pregnant women, those with substance abuse disorders or mental health patients were excluded....." Can a scientic conclusion be based on a survey involving a significant number of laypersons and professionals who may or may not have a good knowledge about the scientific evidence base on the research question? Can we really come to a concluding statement on inclusion and exclusion criteria of cannabis research by a survey?

A significant part of the conclusions drawn by the authors seems to lack any scientific basis. The authors concluded that “given this stakeholders input, we believe there is sufficient support for proceeding with medical cannabis research as an adjunct to analgosedation in critically ill ventilated patients”. This statement seems to lack sufficient ground because the decision to proceed with medical research does not depend upon the lay person’s understanding or viewpoint on a scientific matter but on the basis of the level of evidence and other ethical and operational issues. The authors also conclude that the next step in the research is to determine the correct dosages of medical marijuana also do not seem to be the next step following this survey.

Reviewer #4: #This is a very interesting subject.

#This manuscript can at times be overly descriptive and statistical methods be difficult to understand for a reader who is not well versed in it.

#But, over all this is a well written manuscript and helps bridge a gap in medical literature.

7. PLOS authors have the option to publish the peer review history of their article (what does this mean?). If published, this will include your full peer review and any attached files.

Reviewer #3: No

Reviewer #4: No

---

## [Author Response · Author response to Decision Letter 1]

10 Feb 2021

PONE-D-20-16468R1

Elicitation of stakeholder viewpoints about medical cannabis research for pain management in critically ill ventilated patients: A Q-methodology study

PLOS ONE

Editor 

Thank you for taking the time to review our manuscript

Q1) How generalizable are our conclusions given the sample was derived from a single hospital site?

We specifically chose a Q-methodologic approach because of the limits on sampling from a single hospital site. Unlike other survey techniques that require random sampling to derive a valid representation of the population, Q-methodology only requires that the sample of viewpoints generated (q-set) from the participants (and other sources) to the topic statement represent the theoretic population of all viewpoints. The subsequent ranking of these viewpoints might certainly differ between populations as we discussed in the limitations (lines 345-353), but the underlying theory supporting this model suggests that human beings are much more similar than they are different. There have been q-methodology studies where there was a single participant who provided meaningful and generalizable insights about a topic. As we explained in our Q-methodology design section, the sample sizes needed for this study design are minimal, rarely ever exceeding 40 patients in the literature. The advantage of having used this approach is that we can simply re-administer the q-sort to different populations as part of a qualitative arm of a randomized controlled trial exploring the impact of medical cannabis on analgosedation in critically ill patients. This would provide us some insight on both issues of recruiting and acceptability at different participating test sites.

Reviewer#3

Q1) Strategic sampling of q-statements and participants

In every q-methodology design, there is always an ‘informal’ approach to the selection of the final q-statements from the q-set generated in phase 1. We have included several statements in the limitations section to reflect this intrinsic aspect of every q-study (lines 336-345). As for participant sampling, we wanted to sample participants who might be direct stakeholders in any subsequent study involving medical cannabis in critically ill patients. That was the strategic component of our sampling (lines 140 to 141). Our sampling technique is consistent with that recommended for all q-studies. As for participants completing both phase 1 and phase 2, we specifically asked participants to complete only one or the other, but even if they completed both phases, there is nothing intrinsically wrong with this in q-studies. Many q-studies use the same participants for both phases as there is no strong methodologic reasons for prohibiting this and it is much more efficient. 

Q2) Selection of q-sort statements

We have included more detail about the informal process used in this study to select the final q-sort statements (lines 156 to 169). The final q-sort statements have been included in the supplement Table 1 (lines 168-169). We have also included a statement in the limitations section about the possible selection bias that is intrinsic to q-studies regarding the final inclusion of q-sort statements (lines 336-344). 

Q3) Healthcare providers and lay public nomenclature

We have defined what we mean by these terms and abbreviations used in Table 1 (lines 239 to 244). We did describe the differences in factor loadings between healthcare providers and lay people, including differences between age groups (lines 272 to 282).

Q4) Stakeholders definition

We have included the definition of our use of the term stakeholder in this study (lines 140 to 141).

Q5) Conclusions from viewpoints about subsequent study design elements

In phase 2, each participant was asked to provide qualitative supporting statements about why they had ranked q-sort statements at either extreme end, neutrally and their overall impression of the topic statement. Needless to say, there were many hundreds of these qualitative statements that we could not practically include in the paper. These qualitative statements are an integral part of the q-study conclusions and the reason q-studies are referred to as mixed methods. Many of the conclusions we include are informed by the content of these qualitative statements as much as they are informed by the quantitative factor analysis. As you might imagine, the most difficult part of any q-study is to integrate the factor analysis and the qualitative statements into summary statements as we have done. As for your question about the public participating in research design, this is already happening globally (James Lind Alliance https://www.jla.nihr.ac.uk/, patient-centered outcomes research institute https://www.pcori.org/ to name just a few) and is considered to be the ideal. How we include the public in guiding research can be done in many ways. Whether they sit on research committees or they participate in other ways such as having their opinions or viewpoints know through surveys or other methods seems to be less important than ensuring their opinions and viewpoints are acknowledged and used to help inform research that is patient centered. Our study represents a novel way to include stakeholders and the public into the research planning process.

Q6) Next steps for research based on our study

We sought to answer the following questions: would medical cannabis research be acceptable to stakeholders directly affected by its use in critically patients, and what design considerations should we consider. We feel this q-study helped answer those questions. We do believe a pilot study assessing medical cannabis dosing and safety and feasibility of recruitment are the next steps for medical cannabis research in the critically ill. 

Reviewer#4

There did not appear to be any follow-up questions from this reviewer.

---

## [Decision Letter · Decision Letter 2]

1 Mar 2021

Elicitation of stakeholder viewpoints about medical cannabis research for pain management in critically ill ventilated patients: A Q-methodology study

PONE-D-20-16468R2

Dear Dr. DiDiodato, 

We’re pleased to inform you that your manuscript has been judged scientifically suitable for publication and will be formally accepted for publication once it meets all outstanding technical requirements.

Kind regards,

Saeed Ahmed, MD

Academic Editor

PLOS ONE

---

## [Editor Report · Acceptance letter]

10 Mar 2021

PONE-D-20-16468R2 

Elicitation of stakeholder viewpoints about medical cannabis research for pain management in critically ill ventilated patients: A Q-methodology study 

Dear Dr. DiDiodato:

I'm pleased to inform you that your manuscript has been deemed suitable for publication in PLOS ONE. Congratulations! Your manuscript is now with our production department. 

Kind regards, 

on behalf of

Dr. Saeed Ahmed 

Academic Editor

PLOS ONE